# Towards Understanding Behaviour and Emotions of Children with CLN3 Disease (Batten Disease): Patterns, Problems and Support for Child and Family

**DOI:** 10.3390/ijerph19105895

**Published:** 2022-05-12

**Authors:** Aline K. Honingh, Yvonne L. Kruithof, Willemijn F. E. Kuper, Peter M. van Hasselt, Paula S. Sterkenburg

**Affiliations:** 1Faculty of Behavioural and Movement Science, Vrije Universiteit Amsterdam, 1081 BT Amsterdam, The Netherlands; p.s.sterkenburg@vu.nl; 2Special Education Visually Impaired Children, Bartiméus, 3703 AJ Zeist, The Netherlands; ykruithof@bartimeus.nl; 3Department of Metabolic Diseases, Wilhelmina Children’s Hospital, University Medical Centre Utrecht, Utrecht University, 3508 AB Utrecht, The Netherlands; w.f.e.kuper@umcutrecht.nl (W.F.E.K.); p.vanhasselt@umcutrecht.nl (P.M.v.H.)

**Keywords:** behaviour, emotion, Juvenile Neuronal Ceroid Lipofuscinosis, support, Batten disease, quality of life

## Abstract

The juvenile variant of Neuronal Ceroid Lipofuscinosis (CLN3 disease/Batten disease) is a rare progressive brain disease in children and young adults, characterized by vision loss, decline in cognitive and motor capacities and epilepsy. Children with CLN3 disease often show disturbed behaviour and emotions. The aim of this study is to gain a better understanding of the behaviour and emotions of children with CLN3 disease and to examine the support that the children and their parents are receiving. A combination of qualitative and quantitative analysis was used to analyse patient files and parent interviews. Using a framework analysis approach a codebook was developed, the sources were coded and the data were analysed. The analysis resulted in overviews of (1) typical behaviour and emotions of children as a consequence of CLN3 disease, (2) the support children with CLN3 disease receive, (3) the support parents of these children receive, and (4) the problems these parents face. For a few children their visual, physical or cognitive deterioration was found to lead to specific emotions and behaviour. The quantitative analysis showed that anxiety was reported for all children. The presented overviews on support contain tacit knowledge of health care professionals that has been made explicit by this study. The overviews may provide a lead to adaptable support-modules for children with CLN3 disease and their parents.

## 1. Introduction

CLN3 disease (also known as Batten disease or Neuronal Ceroid Lipofuscinosis (NCL) type 3) is a neurodegenerative disorder of childhood onset. Every year, around 1 per 100,000 children are born with NCL [1], of which type 3 is one of the four primary forms [2], also referred to as the most common type of NCL [3]. In the Netherlands, about 30 children suffer from this disease [4]. CLN3 disease is caused by bi-allelic mutations in the *CLN3* gene, encoding CLN3, a presumed lysosomal protein of a yet unknown function [5,6,7,8]. Without proper functioning of CLN3, ceroid lipopigments accumulate in all body cells, predominantly affecting the retina and the brain [9].

CLN3 disease has a progressive course increasingly damaging the nervous system. The typical clinical course of CLN3 disease involves vision loss at an age of 5–7 years [5,10], followed by epileptic seizures, (Parkinson-like) motor decline, (dementia-like) cognitive decline and behavioural disturbances [11,12], although there is variability in the temporal order of symptom onset and rate of disease progression [13]. Adaptive skills and cognitive function are significantly negatively correlated with disease duration [10,14]. Currently, no treatment can cure or slow down the pace of the disease. In CLN3 disease, treatment is symptomatic and supportive [15]. This means that eventually, children with CLN3 disease lose all communication skills and become completely bedridden. Generally, they do not become older than 23 years.

Research on behaviour in children with CLN3 disease reports emotional and behavioural problems in 50–75% of the children [5,16,17]. Studies are consistent in the finding that there is a higher level of these problems in this group than in peers with typical development [10,16,18,19]. Abnormalities in emotions and behaviour have an increase in puberty and a decrease during the later years of life. These include behavioural problems such as verbal and physical aggression towards others, self-injury and aggression towards objects and emotional reactions such as anxiety, depression, passivity and excessive sadness. In addition, psychotic symptoms, particularly visual hallucinations and delusions, and obsessive–compulsive symptoms have been reported [16,20,21]. Due to the devastating disease course and associated emotional and behavioural problems CLN3 disease has an enormous impact on the quality of life of the child and their family (e.g., parents and siblings) [17,22,23].

During the course of approximately 15 years of the disease, parents of a child with CLN3 disease have to deal with the medical conditions, disturbed behaviour and emotions of their child, as well as their own emotions [20]. Parents of children with health problems have a substantially greater chance of reporting chronic conditions, activity limitations, poor general health, and symptoms of depression than caregivers of healthy children [24,25,26]. In addition, they must learn to deal with these problems that occur more often in children with a chronic illness than in healthy children [27]. Progressive diseases such as CLN3 increase the demands on the family system as the child’s functionality decreases and the need for care increases [28,29,30]. Caring for a child with CLN3 disease places a very significant burden on the family system and includes parental feelings of loss, impact on family relationships and lack of understanding within the health/social insurance systems. While taking care of a child with deteriorating functions, they must also struggle to ensure that the child receives the supports and care they need from the health and social care system [31].

Exactly how CLN3 disease leads to specific behaviour and emotions is not yet fully understood. Research shows that both internalizing and externalizing symptoms are clinically elevated in children with CLN3 disease. The internalizing symptoms are significantly related to parents’ self-rated quality of life and may provide input for further study and intervention [16]. In the current study, we want to further explore the behaviour and emotions of children with CLN3 disease.

Although support programs have been developed for children with behavioural problems, behavioural interventions have not been specifically evaluated for children with CLN3 disease [16]. Children with a decline in memory and attention may have limited abilities to benefit from behaviour management programs [32]. An educational and social support program has been developed for children with CLN3 disease, including for example music therapy and braille education [33]. However, no behavioural support or a formalisation on manners or contact with respect to these children has been formalised. In the Dutch NCL centre of expertise (a collaboration between Bartiméus, Royal Visio and the Wilhelmina Children’s Hospital) children with CLN3 disease receive medical care, counselling and acquire additional therapy (e.g., speech therapy, physiotherapy and behavioural support). Parents of these children receive support as well. Therefore, health professionals working with these children and their parents have a lot of knowledge on specific CLN3-support. This study aims at making this knowledge explicit.

## 2. Materials and Methods

### 2.1. Participants

Patient file information of eleven children with a genetically confirmed diagnosis of CLN3 disease was included in this study. These children did not participate actively: only information previously obtained was used. The children, four girls and seven boys, received care through the NCL expertise centre at Bartiméus, where over the years patient files were created and stored. Nine of the patients were identified with the classic form of CLN3 disease, starting with vision loss between the age of 5–10 years, followed by loss of motor coordination, mental decline and seizures from the age of 10–12 years. Two of the patients were identified with the protracted form of CLN3 disease, associated with a slower disease progression [34]. The patient files contain information concerning the child’s disease from medical specialists (diagnosis, follow-up controls, referrals), developmental psychologists, teachers and care professionals. This information was supplemented with information from previously conducted interviews with parents of a child with CLN3 disease [23] and quantitative data from the Unified Batten Disease Rating Scale (UBDRS) [35]. The UBDRS is a questionnaire targeted on CLN3 disease patients, used periodically to monitor physical and cognitive abilities as well as behaviour and emotions. The UBDRS is a valid and reliable rating scale for assessing the severity and rate of progression of CLN3 disease [18,35]. For seven children, UBDRS data were available. In addition to information about these children, the interviews provided information about the parents, especially considering their emotions and way of living. For seven children, interviews with both parents were available. Table 1 provides an overview for each participant of the three sources of information: patient file, interview parents and UBDRS-data.

Parents of children with CLN3 disease were sent an information letter in which they were asked whether the patient files of their child including the UBDRS-data could be used for analysis in this study. Parents who had been interviewed about the disease of their child in an earlier study, were approached separately with the same question. Patient files, interviews and UBDRS-codes for which parents returned their informed consent letter, were anonymized before the analysis started. The Scientific and Ethical Review Board (VCWE) of the Faculty of Behaviour and Movement Sciences, Vrije Universiteit Amsterdam (VCWE-2020-078) provided their ethical approval for conducting this study.

### 2.2. Procedure

A qualitative research approach is useful for developing knowledge in complex research areas by applying an explorative design [36]. In order to acquire a better understanding of the behaviour and emotion of children with CLN3 disease the qualitative approach was used in this study. To also account for our secondary goal, to acquire an overview of the support that these children and their parents receive, a Framework analysis [37,38] was used for the analysis of the qualitative data, where an analytic framework or coding manual was developed as a tool for analysis. The Framework method produces structured output of summarized data [37] which, in this case, can lead to overviews (summaries) of problems and support with respect to CLN3 disease. The analytic procedure consisted of the following. As a first step, the patient files and interviews were read multiple times for familiarization with the data. Open coding was applied to the data, where words or sentences that were or could be related to (1) behaviour and emotions of the child with CLN3 disease and (2) support of the child or the parents, were labelled with codes. Thereafter, axial coding was used, where codes were compared, and categories could emerge. This process resulted in a working coding manual. During the process of coding, the working coding manual was updated in an iterative process and evaluated after every document, by comparing the new added codes among researchers [39]. Differences in codes were discussed and resolved through consensus. In this way, the working coding manual evolved from a dynamic document into a static hierarchical coding document, including main codes and subcodes, resembling the topics derived from the data, linking to behaviour and emotions of children with CLN3 disease. The decision of changing the working coding manual into a static coding manual was based on saturation after approximately coding one third of the data: hardly any new codes were added to the manual at that point. The coding manual provides the analytic framework for analysis, containing major themes and concepts, and can be found in the Appendix A, Table A1.

The coding manual was developed by three researchers (A.K.H., Y.L.K. and W.F.E.K.). After finalizing the coding manual, a master-degree research assistant was trained to take part in the coding process, where the interrater reliability was used to assess the performance [40]. For this training, the coding manual was presented together with sample data. The reliability was checked by calculating a percentage of interrater reliability where the codes of the developing team were used as ground truth. After reaching a reliability percentage of 80% in practice rounds, the independent coder started to take part in the actual coding process.

The three researchers (A.K.H., Y.L.K. and W.F.E.K) and the independent coder coded all qualitative data independently. In addition, the data previously coded with the working code manual, was recoded with the finalized coding manual. The coding process consisted of two stages: First, segmenting the data into meaningful pieces or data units, and second, labelling each unit with a code [39,40]. A data unit is a piece of text on a specific topic. For example, the text “She was suddenly very sad about her curly hair” can be labelled with the main code ‘emotional factors’ and subcode ‘sadness’. The two stages of segmenting and labelling are interrelated, as a data unit consists of enough text to understand the topic, but at the same time should be as short as possible and is preferably labelled with only one code. Since this appeared not always to be possible (a single sentence might cover two different topics), double codes were occasionally used as well. For each client, the data units were listed in an Excel file. Each data unit was associated with multiple columns: the name of the patient file or interview it came from, the name of the professional who wrote it, the date, the age of the client, the stage of the disease [41], and the labels (one or two main codes and subcodes) that were given by the coder. For the double coded data, only one of the coders segmented the data. Thereafter, the data units were independently coded by the two coders [40].

### 2.3. Data Analysis

All qualitative data (the patient files and interviews) were coded, one third was double coded by two coders. The mean interrater reliability was calculated according to the percentage of agreement. (Cohen’s kappa could not be used, because a piece of text could be given more than one main code). The percentage of agreement on the main codes on the double coded part of the data was 79%. The differences in codes were resolved through discussion until consensus was reached. The rest of the qualitative data was coded by a single coder. If this coder had doubts on the code to be given, this was discussed with one of the other coders until consensus was reached. In total, 16% of these remaining codes were discussed.

The coded qualitative data (patient files and interviews) were analysed in two stages. In stage one, the data were summarised per client, following the main codes of the coding manual. For each client, the data units were ordered chronologically per main code, resulting in an overview of items describing these topics for this child. In this way, for every main code (for example ‘emotional factors’ or ‘behavioural factors’) the development of specific topics among the subcodes (for example ‘anxiety’ or ‘aggressiveness’) could be read from the chronological text. The UBDRS-data were used to complement the qualitative data for the clients it applied to. The reported emotions and behaviours from these data were added to the chronological description.

The framework analysis approach gives rise to different types of results. Summarised data can give an overview of a particular main code and connections between main codes can be investigated to look for explanations or patterns. Below, questions giving rise to an overview are marked with a ‘•’ and those marking connections between main codes are marked with a ‘+’.

From the analysed files for individual clients, answers to the following questions could be extracted:Which emotional and behavioural symptoms occur?
+How do these symptoms develop?+Which circumstances or events that lead to (solutions of) these symptoms are seen?+Are there possible protective factors in the system around the child?
What kind of support do the child and the family receive?With what kind of problems do the parents have to deal?

In stage two of the analysis, the individual analyses were combined, to answer the following overarching questions:What categories of emotional and behavioural symptoms are found?
+Are there typical combinations of emotional and behavioural symptoms?+Is there a typical development of emotional and behavioural symptoms?+Are there typical circumstances or events that lead to (beginning and ending of) symptoms?+Are there commonly identified protective factors in the system around the child?
What categories of support that the child with CLN3 disease receives can be found?What categories of support that the parents of a child with CLN3 disease receive can be found?How can the problems that these parents deal with be categorised?

The answers to the questions marked with a ‘•’ now give rise to overviews or summaries of symptoms and support for children with CLN3 disease and overviews of problems and support for their parents. The answers to questions marked with a ‘+’ give rise to explanations or patterns. The analyses were performed by two researchers (A.K.H. and Y.L.K). Thereafter, the results were discussed in a larger group (A.K.H., Y.L.K., W.F.E.K. and P.S.S.) until consensus was reached.

As a first step into the quantitative analysis, the UBDRS-data were organised using tables and visualised using graphics to receive an impression of (1) the development of emotional and behavioural symptoms and (2) of the prevalence of these symptoms in this set of clients.

## 3. Results

For the quantitative analysis, the UBDRS-data were used. However, since UBDRS-data of only seven clients were available, only basic statistics could be used. In Appendix A, an overview of the data is presented (Table A2). The UBDRS-data show that the seven clients for whom UBDRS-data was available are all dealing with anxiety (Appendix A, Table A3). Some emotional and behavioural symptoms increase in frequency and intensity and some decrease (Appendix A, Figure A1). No typical pattern (such as an n-shape, observed by Adams et al. [16]) was found.

Below, the results of the qualitative analyses are presented, following the order of the overarching questions listed in the previous paragraph.

An overview of behavioural, emotional and psychological symptoms resulted from the analysis, together with examples, where eleven distinct categories could be observed (See Table 2). Only instances of symptoms where an example could be given (the client number is given within parentheses), were listed. This means that the UBDRS-data in which only the prevalence of certain symptoms was indicated, have not been used here.

To identify typical combinations and development of symptoms for each client, all behavioural and emotional symptoms were listed in chronological order and compared to symptoms of other clients. In this way, combinations of occurring symptoms, as well as typical sequences of symptoms: two or more (particular) consecutive symptoms that occur with several clients, could be identified. However, no typical combinations of symptoms and no typical development of symptoms could be identified. Every client showed a unique situation.

To identify events or situations leading to (beginnings and endings) of behavioural and emotional symptoms data was investigated to see whether these symptoms were cooccurring with other information, such as physical symptoms of CLN3 disease. However, not all cooccurring pieces of information are necessarily connected to each other. Two cooccurring items were marked as a connection if these items started at the same time. Additionally, if a relation between two items was mentioned in the patient file or in an interview by either the parents or a health professional, this was notated as a connection as well. Note that, because of this method, no slowly progressing connections could be found. Only results accompanied by a substantial change at a certain moment are shown. In this way, for some clients, a situation or event leading to a beginning or ending of a certain behaviour or emotion was identified. These are listed below. From the list, no typical situations or events leading to behavioural and emotional symptoms could be identified.

Observed connections between CLN3 disease symptoms and emotions:*deteriorating vision -> anxiety*For client 7 the anxiety starts at the age of 6, when also his vision deteriorates rapidly.*declining mobility -> anxiety*The increase of anxiety of client 1 starts at the same time as a fast decline in mobility. In situations where she is standing or walking without support, she shows fear of falling.*decline in cognitive skills (+rivalry) -> anger*As a result of the decline in cognitive skills of client 11, and the (normal) cognitive development of her younger sister, the two sisters are at the same cognitive level at some point, which leads to great rivalry and therefore great anger of client 11. When, half a year later, the younger sister has overtaken her older sister on cognitive skills, the anger and tantrums of client 11 fade away.*decline of language skills -> anger*If client 4 cannot make clear what he wants to say and other people do not understand him, he becomes angry and agitated and raises his voice to repeat what he is saying, over and over again.*physical decline -> sadness*Client 6 goes through periods of sadness because of the things he cannot do anymore and cries at night about this.*cognitive decline -> sadness*Client 11 cannot remember the names of certain things, resulting in other people not understanding what she means. This makes her sad.Observed connections between CLN3 disease symptoms and behaviour:*decline in motor skills and energy -> passiveness*Due to the decline in motor skills, client 8 can do less and less things independently. She becomes increasingly passive.*decline in energy -> less rebellious*Around the age of 20, client 4 acts less rebellious as a consequence of a decline in energy.

One could think that commonly identified protective factors may exist, preventing a child with CLN3 disease from developing behavioural or emotional symptoms. However, from our analysis, no protective factors emerged. The severity of the behavioural symptoms does not seem to coincide with environmental or situational factors, such as, for example, the stability of the family. Illustrative of this is the fact that our data contained information on two brothers (clients 3 and 6) who both have CLN3 disease, one of whom has many behavioural symptoms while the other has not.

The support that children with CLN3 disease receive can be divided into several categories. There is practical support, for example from physiotherapists and behavioural therapists; secondary support, consisting of advice from health professionals to teachers and parents; and (behavioural) medication. Furthermore, seven categories of specific CLN3 support could be identified that are used when being in direct contact with children with CLN3 disease, concerning the visual and cognitive decline faced by the children:Expressing in words what the environment looks like and what is happeningOffering structure: clear daily routines, rules and boundariesOffer extra explanation and attentionClarify communication and support social contactHelp to memorize things and to keep memories aliveStimulation and encouragementAcknowledging emotions and helping to control them

The observed support parents receive, could be categorised into four categories, displayed in Table 3.

Finally, an overview of the problems that parents run into was made, see Table 4. The first four categories of this list: ‘emotions, interactions with the child, future prospect and practical issues’, correspond with the items in Table 3, which means that although these problems were mentioned, support concerning these problems was also mentioned. The bottom four categories consist of problems which were not mentioned in combination with support.

## 4. Discussion

In this study, data of children with CLN3 disease were analysed using a primarily qualitative approach. The focus was to examine the behavioural and emotional experiences, and support for the child and family. The results list the behavioural, emotional and psychological symptoms these children show. Additionally, overviews were made of the support these children and their parents receive. Furthermore, the problems that these parents face were listed. The analysis showed that for some children, certain behavioural and emotional symptoms are a direct consequence of CLN3 disease symptoms. The examples given in the qualitative analysis illustrate the findings. Finally, anxiety was noted for all clients whose UBDRS-data was available.

The connections found (for some children) between emotions and behaviour and the CLN3 disease symptoms varied, and no general conclusions could be drawn. Cognitive decline could, for example, lead to anger but also to sadness, and physical decline could lead to sadness but also to anxiety. This means that, according to our analysis, every child reacts in a personal way to symptoms of CLN3 disease. It may be possible that the dataset was not big enough and too few clients were analysed to show significant behavioural and emotional patterns. Future research could shed light on this question. However, the present results are in line with research by Von Tetzchner et al. [17] who conclude that children with CLN3 disease react to their difficult life situation in diverse ways, both emotionally and behaviourally.

The categories for the behavioural, emotional and psychological symptoms found in this study differ slightly from the standard UBDRS-categories ([18] See also Table A2 in the Appendix A), since a data-driven approach was used where the categories emerged from the data. One of the categories, ‘passive’ is close to the UBDRS-category ‘apathy’ but differs in the sense that the passive behaviour did not necessarily include indifference and lack of emotion, which are implied by the term apathy. ‘Stereotype behaviour’ and ‘compulsion’, which are separate categories in the UBDRS, blended in our data-analysis. Four extra categories emerged from our analysis: ‘rebellious behaviour’, ‘demanding behaviour’, ‘focussing on the familiar’ and ‘increased sensitivity’.

From the connections between CLN3 disease symptoms and emotions, and from the fact that anxiety was noted in all UBDRS-files, it can be concluded that emotions play a key role in the behaviour of these children. This is in line with Adams et al. [13] who found correlations between some emotions and behaviour. Possibly, typical behavioural of children with CLN3 disease may come from different causes: first, behaviour that can be seen as a direct consequence of the deterioration of certain brain functions, such as uninhibited behaviour, and second, behaviour that is a consequence of certain emotions, for example anxiety, that could lead to behaviour to look for control when one is slowly losing control, such as ‘focussing on the familiar’ and ‘demanding behaviour’. It is known that emotions can sometimes act as mediator between stimuli and behaviour [42] and the question is whether that is the case here. Future research is needed to investigate the connection between emotion and behaviour in CLN3 disease and the question whether a distinction can be made, dividing behaviour based on its cause. Results may contribute to better behavioural support for children with CLN3 disease.

While educational and social support modules for children with CLN3 disease have been developed and focus for example on music therapy and braille [33], no list of advice on direct contact with these children has been formalised. With the list of seven categories of specific CLN3 support, tacit knowledge of professionals working with children with CLN3 disease has been made explicit. This list may form the beginning of a more formalised adaptable support-module, and professionals working with children with CLN3 disease, such as teachers and therapists, can use this list as a guide.

The eight categories of problems of parents describe the different problems parents face from the moment that they are confronted with the diagnoses of their child. The families taking part in this study included children living at home as well as children part-time living in a specialized home care facility. These perspectives may give rise to different parental problems both included (however blended) in Table 4. These categories of problems can be compared with the themes resulting from the study by Krantz and colleagues [31] on parental experiences of having a child with CLN3 disease. Their theme ‘recurring losses’, referring to the feeling of losing a healthy child, the child’s loss of abilities and the loss of relationships is close to our category ‘emotions’ that includes sadness, anger and despair resulting primarily from these losses. Their theme ‘disruption of the family system’ relates to our category ‘interaction with other people’, since both include the changed relation with (and attention for) other children in the family and romantic relationships that are under pressure. Finally, the theme ‘Society is not developed for a progressive disease’ included difficulties parents faced with respect to contact with the health system, which is described in our category ‘interaction with health professionals’. Comparing our categories to another study, on the psychosocial impact of parenting a child with a lysosomal storage disorder [43], overlap can be found as well. In this study, two themes, ‘uncertainty and the unknown’ and ‘finding a way forward’ are both related to our category ‘future prospect’, while the theme ‘finding a way forward’ is also related to our category ‘choices with respect to CLN3 disease’. Their theme ‘all-encompassing impact’ includes subthemes ‘behaviour’, ‘emotions’ and ‘physical demands’ which correspond to our categories ‘interaction with the child’, ‘emotions’ and ‘practical issues’, respectively.

Despite all problems parents with a child with CLN3 disease face, no official support module has been developed. However, parents with a child with CLN3 disease do receive support and this study has made the different kinds of support that parents receive explicit. The categories of support for parents are outlined in Table 3. It is worth noticing that three of the four categories include examples of contact with other parents with a child with CLN3 disease, either via the patient association or direct contact. This is in line with the study by Cozart et al. [22] showing that parents with a child with CLN3 disease expressed an interest in and preference for parent-to-parent communication for support and information about the disease. By comparing the categories of support with the categories of problems, it becomes clear that for some problems support is difficult to organise or has not been developed. The support categories outlined in Table 3 and problem categories from Table 4 may serve as point of departure for a support-module for parents with a child with CLN3 disease. The fact that many differences exist between children with CLN3 disease, indicates that case-sensitive or adaptable modules are needed.

A limitation of the study was the non-uniformity of the data with respect to age and incompleteness of the data. In addition, the amount of data per client varied widely. For some clients, many reports were included in the patient files, while for others, only a few reports were available. The same was true for the UBDRS data: no structural periodic measurements were carried out for all clients. Furthermore, no perfect agreement between the UBDRS data and patient files of clients in a specific period was present. For example, if anxiety was reported in the UBDRS, one would expect to find anxiety also reported in the patient file of the same period, but this was not always the case. It can be concluded that relevant information is still missing, so certain connections (e.g., typical combinations of/connection between symptoms) were not found. Due to missing information in the patient files, the different stages of the CLN3 disease could also not be identified. CLN3 disease can be divided into four stages where the boundaries between the stages are marked by moments of increased physical deterioration [41]. In this study it was not possible to identify typical behaviour or emotions linked to a certain stage of the CLN3 disease. It is important to monitor the stage of disease together with the UBDRS, so more can be learned about behaviour and emotions regarding the stage of the disease and to offer specialized care and support. Since many differences exist between children with CLN3 disease, we recommend working towards case-sensitive or adaptable modules.

Finally, although the experiences of siblings of children with CLN3 disease were not investigated in this research, from the interviews it became clear that many siblings face similar challenges as the parents. Therefore, in future research on support programmes for families, siblings deserve distinct attention.

## 5. Conclusions

CLN3 disease is a rare neurodegenerative disease causing not only vision loss and decline in cognitive and motor skills, but also typical behaviour and emotions. Decline in physical and cognitive skills give rise to certain emotions and behaviour. These behavioural and emotional symptoms are different for each individual, although general characteristics can be found. More research is needed to examine the connection between emotions and behaviour. The examples resulting from the qualitative analysis illustrate the symptoms and problems that these children and families deal with and the support that they are currently receiving. The overviews may provide a lead to support-modules for children with CLN3-disease, focusing on behaviour and emotions, and for the families as a whole.

## Figures and Tables

**Table 1 ijerph-19-05895-t001:** Available data per participant.

Client Number	Gender	CLN3 Phenotype	Patient File (Age Range)	Interview Parents	UBDRS (Number of Periodic Measurements)
1	f	classic	yes (6–20)	yes	yes (10)
2	m	classic	yes (13–23)	no	yes (5)
3	m	classic	No	yes	yes (6)
4	m	classic	yes (8–22)	yes	yes (8)
5	m	protracted	yes (6–14)	yes	yes (7)
6 (brother of 3)	m	classic	No	yes	yes (2)
7	m	classic	yes (6–13)	no	yes (1)
8	f	protracted	yes (6–22)	yes	no
9	m	classic	yes (7–10)	no	no
10	f	classic	yes (6–18)	no	no
11	f	classic	no	yes	no

**Table 2 ijerph-19-05895-t002:** Observed behavioural, emotional and psychological symptoms in children with CLN3 disease.

Symptoms	Clients + Examples
Anxiety	Several examples of anxiety were found, some in relation to motor skills (1—afraid to fall) or vision (1—unexpected sounds), and others more general (5—fear for unknown things, 7—afraid at night or fear in the playground).
Sadness	Sadness about failing communication skills (11) and sadness about physical deterioration (6) was noticed. Delayed sadness, where the emotion comes only the next morning was repeatedly found for one client (1). Old songs that one client used to sing along with, make her sad at later age (1).
Rebellious	At home, some clients explore their boundaries (9) or get angry quickly (10). Not being understood was found to give rise to angry behaviour (4), as well as having to do things that one does not want to (8).
Aggression	Yelling and screaming was found to occur when getting angry or frustrated (1, 3); sometimes aggression towards others (1—pushing others), but also to the self was reported (1—hits herself when having an episode). Aggression towards things was also seen (3—trying to destroy things). In one case, rivalry between sisters (the younger sister levelling the cognitive skills of a client at some point) led to aggression (11).
Demanding	Demanding and claiming behaviour was seen towards parents and siblings (5, 8, 10) as well as dominant and demanding behaviour in general (4).
Focussing on the familiar	Several clients have a preference for familiar things, activities and rituals (1, 4, 5, 7, 8), including familiar topics in conversations (4).
Obsession	Several obsessions were reported, like obsessions with hair (1), soccer (2), ‘Sinterklaas’ (3), clothes with pictures of horses on it (11), stones and steam engines (4).
Stereotypical and compulsive behaviour	Several clients have lost inhibition with respect to talking (1, 2, 4, 5, 6, 8). They can go on and on about things and focus on the specific topics of their interest. Inappropriate behaviour in puberty was also noticed (4—touching ‘accidentally’ women’s breasts), as well as stubbornly continuing with something that is not possible anymore (11—typing letters to a friend).
Passive	Taking less initiative, preferring to listen while others talk and making less contact is seen with clients in a later stage of the disease (1, 2, 8), while phases of passiveness have also been noticed earlier on (need a lot of stimulation to get to an activity—10).
Increased sensitivity	At a certain point during the disease, some clients become more sensitive for noises or (mild) chaotic situations, like birthday parties (4, 8, 11). They get overstimulated and may even get a panic attack (8).
Hallucinations	For some clients, delusions and auditory hallucinations start during a later stage of the disease (2, 8), while one client was said to always have been in a fantasy world part of the time (3).

**Table 3 ijerph-19-05895-t003:** Observed support for parents.

Type of Support	Examples
Emotional support	from partner, family, friends, health professionals, patient association and religion
Support for interaction with their child	advice from health professionals and other NCL-parents on providing structure, stimulation, explaining things, special toys, etc.
Psycho education—learning about CLN3 disease	explanation on (future) NCL-symptoms and how to deal with those, from health professionals and patient association
Practical support	e.g., adjustments in the house, nurses at home, financial advice, care facilities

**Table 4 ijerph-19-05895-t004:** Observed problems of parents concerning CLN3 disease.

Type of Problems	Examples
Emotions	Feelings of anger, sadness, despair, fairness and depression
Interaction with their child	How to react to strong emotions and problematic behaviour of the child? Difficult to stay patient. What to say when child asks about future?
Future prospect	(No) desire to know about future prospect; prepare for future.
Practical issues	Problems to arrange wheelchair, stairlift, parking permit, etc.; financial cuts at care facilities.
Energy	The care takes a lot of (physical and mental) energy and sacrifices
Choices with respect to CLN3 disease	e.g., choices between sheltered care facilities or care for the child at home, tube feeding, euthanasia, etc.
Interaction with health professionals	Disagreement with health professionals, second opinions, frustration about communication and waiting (lists)
Interaction with other people	Less attention for other children, relationship under pressure, losing friends because of limited time, harsh judgements from (ignorant) people

## Data Availability

Data is available on request.

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
