# Peer review of "Towards Understanding Behaviour and Emotions of Children with CLN3 Disease (Batten Disease): Patterns, Problems and Support for Child and Family"

_ijerph, 2022, doi:10.3390/ijerph19105895_

Round 1
Reviewer 1 Report
The study makes an ambiguous impression. The object of the study is patients with a congenital incurable disease with an irreversibly unfavorable prognosis. There are very few such patients, but the living of each of them in families is a big problem for others. At the same time, such patients may not be in families, but in specialized medical institutions, and then the nature of the problem indicated by the authors changes greatly. The development of the disease is characterized by the appearance of mental and emotional abnormalities in patients, as well as violations of their functional and physical condition. In all patients, certain features in the development of these abnormalities are revealed, with a general clinical picture. The main problem of the study is that, in general, the novelty of the study is not clearly and clearly formulated and this is the most important claim to the work. In addition, the authors conducted a rather weak literary discussion of both the results obtained and the state of the problem as a whole - this is clearly evidenced by the number of literary sources used by the authors - only 28. Such a small number of literary sources cited by the authors in relation to the reviewed article raises the question of the real importance of the problem under consideration. In general, the relevance of the study is completely not clearly visible in the work, and this requires its mandatory revision, since the authors in tex are limited only to general statements about the need for some new developments, which they have not been presented due to the high variability of the individual picture of the psychological changes they have identified in patients.
Author Response
Dear reviewer,
We want to thank you for the time that you spent reading our manuscript and providing valuable feedback. We understand your concerns and have tried to strengthen the manuscript along the lines of your comments. We have included extra literature in the introduction and we have tried to explain and motivate our study better. In the discussion we have added a comparison of our results to results from the literature, so as to highlight the novelty of our study. In the manuscript, track-changes have been used to indicate changes.
We agree that the nature of problems change if the child with CLN3 disease is not living with its family but in a specialised care facility. We have included this point in the discussion.
Thank you again for your review.
Reviewer 2 Report
Dear authors "Towards understanding behaviour and emotions of children with CLN3 disease (Batten disease): a retrospective file research",
The study seemed a nice contribution to the literature; however, important amendments have to be addressed before considering publication due to lack of transparency, validity, and reproducibility. The structure of the manuscripts needs to be revised. English needs to be improved throughout (e.g., mixing American and British English), the manuscript would benefit from thorough proofreading by a native English speaker to improve grammar.
Specific comments:
TITLE: "A retrospective file research"? After a search in databases, there are no guidelines or similar references. The title seems not appropriate; therefore, the authors are encouraged to modify it accordingly.
ABSTRACT:
Authors should report/highlight quantitative results in the abstract.
Line 12-13: Could the authors please state the prevalence in the population?
Line 15: To gain a better understanding is not a clear objective. The aim needs to be modified accordingly.
Line 16: Methodology is not clear so please elaborate. Could the authors explain why did not follow a standardized methodology (e.g., STROBE guidelines) to perform the study? This is important under the current paradigm of open science, transparency, and reproducibility of research studies.
Line 19: "overviews"? Are those results of a framework analysis approach? If no validation of the used instruments nor Delphi's methodology was implemented, how do we ensure the reliability of the results?
Line 22-24: The contribution to scientific literature is not clear. Batten disease has been studied since 1986 to support children and families; in fact, "special remedial education, special social support, and other medical, social, and/or vocational services" have been designed by several organizations - check: https://rarediseases.org/rare-diseases/batten-disease/
INTRODUCTION & BACKGROUND
The introduction is missing important citations to support some statements. Also, there is a lack of genetic background. This section seems too long and citations need to be formatted according to the MDPI style.
Lines 29: Please insert the citation.
Line 30-31: Please include information regarding the worldwide prevalence of juvenile CLN3 disease.
Line 32: There are several studies that have evaluated this. Please insert citations accordingly and elaborate on the molecular genetics behind the disease.
Line 33-34: Citation missing here.
Line 45-46: How different is the impact of the juvenile CLN3 disease over other rare conditions? Also, a citation is missing here to support the impact on families (e.g., Krantz et al. 2022).
Lines 47-91: This should be shortened and merged with the previous paragraph. Too much redundant information. In fact, some sections can be relocated to the DISCUSSION section.
Line 104-106: The aim of the study seems to be "to evaluate changes and patterns in patients' behavior for supporting treatment practices and counseling affected families". Authors should revise the title, objectives, and discussion accordingly.
MATERIALS AND METHODS
Authors have performed an observational study (i.e., STROBE guidelines) but also used available information from a questionnaire targeted at juvenile CLN3 disease patients (i.e., CHERRIES guidelines). Please adequate your manuscript's structure to these international guidelines - STROBE (follow strictly the subsections and report procedures accordingly). In fact, there is an extension of the STROBE guidelines focused on genetics-related research: STREGA (PMID 19192942).
Line 109-111: Please elaborate. How was this information collected? Where? IRB approval? Specify procedures.
Line 113-115: Was this open and free data for re-analysis? Did you register the protocol of this study?
Line 116-117: Has been this instrument validated in the Dutch population? Please cite accordingly.
Table 1: Please elaborate on the meaning of "classic" and "protracted".
Line 109-119: This subsection needs to be re-written. It did not contain specific information regarding participants.
Line 122-129: Relocate this information in the corresponding subsections based on STROBE guidelines (STREGA if necessary).
Line 130-152: "Consensus" of co-authors only? How is this valid in the light of the scientific community? Could the authors please explain why did not perform a Delphi exercise or at least a content validity analysis? How did you measure "reliability"?
Line 146-165: These are the second and third steps of the "Framework analytic procedure"? English editing is needed to improve readability. Too long paragraphs make it difficult to understand clearly the performed procedures.
Line 166-174: This should be relocated as part of the "Data analysis".
Line 206-208: Not clear. Please elaborate on the analytical workflow.
Line 208-209: Delete.
RESULTS
This section needs to be improved. Too much text makes it difficult to interpret findings. Please follow STROBE guidelines and support your findings with data from the current study.
Line 211: all clients? They were not explained in the METHODS section. Only 7 clients? Where do they come from?
Line 217-218: Difficult to follow. What do the authors mean by "overview"? This is part of the Framework analysis? If so, how can we as researchers validate the objectivity of the results?
Line 227-228: "However, no typical 227
combinations of problems and no typical development of problems could be identified": How do you support this statement based on your findings? Lack of reproducibility.
Line 230-272: Causation is beyond the scope of this study design. Authors should avoid any attempt to explain causation if any molecular, mechanistic, or robust statistical modeling (even with AI or ML procedures) were performed.
Line 276-279: Please report those correlation outcomes! Please elaborate differences between brothers in terms of the genetics analyses.
DISCUSSION
This section is far from objectivity and is non-supported by the findings of this study based on the lack of reproducibility and validity of the data.
Perhaps a scoping review would be a better form to use the information collected by the authors given the lack of standardization of the current study - which is highly expected and required in a retrospective cross-sectional study.
CONCLUSIONS
These statements should be based on your findings according to the limits of the study aim. Generalities are not accepted as conclusions.
Author Response
Dear Reviewer,
Thank you for your extensive review and useful feedback. We apologize for mixing up American and British English and have corrected this. In addition, the manuscript is read and corrected by a native English speaker. Below, we respond to every specific point of feedback. In the manuscript, track-changes have been used to indicate changes. For the readability of the reviewed document some lay-out, spelling and grammer changes are made without using track-changes.
Specific comments:
TITLE: "A retrospective file research"? After a search in databases, there are no guidelines or similar references. The title seems not appropriate; therefore, the authors are encouraged to modify it accordingly.
We understand that it would be better to use terms that are more commonly used. Therefore, we have changed the title to: Towards understanding behaviour and emotions of children with CLN3 disease (Batten disease).
ABSTRACT:
Authors should report/highlight quantitative results in the abstract.
Line 12-13: Could the authors please state the prevalence in the population?
Line 15: To gain a better understanding is not a clear objective. The aim needs to be modified accordingly.
Line 16: Methodology is not clear so please elaborate. Could the authors explain why did not follow a standardized methodology (e.g., STROBE guidelines) to perform the study? This is important under the current paradigm of open science, transparency, and reproducibility of research studies.
Line 19: "overviews"? Are those results of a framework analysis approach? If no validation of the used instruments nor Delphi's methodology was implemented, how do we ensure the reliability of the results?
Line 22-24: The contribution to scientific literature is not clear. Batten disease has been studied since 1986 to support children and families; in fact, "special remedial education, special social support, and other medical, social, and/or vocational services" have been designed by several organizations - check: https://rarediseases.org/rare-diseases/batten-disease/
We have rewritten the abstract and have tried to take into account your feedback. However, the maximum length of the abstract is only 200 words. Therefore, some of these points of feedback, we have taken up elsewhere in the article. For example, in the abstract, the aim is formulated in one sentence, but in the introduction, we further specify our objectives. The overviews are indeed the results of the framework analysis. Please see our elaboration/clarification in the article and our response to your feedback on guidelines and analyses below. We believe that we have followed a standardized methodology (Qualitative study using Framework analysis) and reported the results following guidelines that are appropriate for this (JARS-Qual). We understand from your words that the contribution to the scientific literature is not entirely clear. You are right to say that several kinds of treatments to support the child and the family has been developed. However, these are mainly focused on supporting/dealing with the physical and cognitive deterioration of the child. The behavioural and emotional problems (some) children show, come on top of this, and need to be understood better before a support module can be developed. With this study, we have tried to make this first step of trying to understand these problems better. With your feedback (about the contribution not being clear) in mind, we have improved the article.
INTRODUCTION & BACKGROUND
The introduction is missing important citations to support some statements. Also, there is a lack of genetic background. This section seems too long and citations need to be formatted according to the MDPI style.
Lines 29: Please insert the citation.
Line 30-31: Please include information regarding the worldwide prevalence of juvenile CLN3 disease.
Line 32: There are several studies that have evaluated this. Please insert citations accordingly and elaborate on the molecular genetics behind the disease.
Line 33-34: Citation missing here.
Line 45-46: How different is the impact of the juvenile CLN3 disease over other rare conditions? Also, a citation is missing here to support the impact on families (e.g., Krantz et al. 2022).
Lines 47-91: This should be shortened and merged with the previous paragraph. Too much redundant information. In fact, some sections can be relocated to the DISCUSSION section.
Line 104-106: The aim of the study seems to be "to evaluate changes and patterns in patients' behavior for supporting treatment practices and counseling affected families". Authors should revise the title, objectives, and discussion accordingly.
We have rewritten the introduction, taking into account your points of feedback. We have strengthened the genetic background, included more literature, removed redundant information and clarified the aim of our study. We agree that aim of the study was not clear enough and did not correspond well enough to the title and abstract. We have revised the title, abstract and discussion accordingly. Thank you for your important feedback on these aspects.
MATERIALS AND METHODS
Authors have performed an observational study (i.e., STROBE guidelines) but also used available information from a questionnaire targeted at juvenile CLN3 disease patients (i.e., CHERRIES guidelines). Please adequate your manuscript's structure to these international guidelines - STROBE (follow strictly the subsections and report procedures accordingly). In fact, there is an extension of the STROBE guidelines focused on genetics-related research: STREGA (PMID 19192942).
We understand the preference for using an international guideline to structure the manuscript and we thank the reviewer for these specific suggestions. Since our study focuses on behaviour and emotions (instead of on epidemiology and genetics), we feel that the JARS-Qual reporting standards are more applicable to our research. We have used these in our manuscript. These standards have been described in:
Levitt, H. M., Bamberg, M., Creswell, J. W., Frost, D. M., Josselson, R., & Suárez-Orozco, C. (2018). Journal article reporting standards for qualitative primary, qualitative meta-analytic, and mixed methods research in psychology: The APA Publications and Communications Board task force report. American Psychologist, 73(1), 26.
Line 109-111: Please elaborate. How was this information collected? Where? IRB approval? Specify procedures.
Line 113-115: Was this open and free data for re-analysis? Did you register the protocol of this study?
Line 116-117: Has been this instrument validated in the Dutch population? Please cite accordingly.
Table 1: Please elaborate on the meaning of "classic" and "protracted".
Line 109-119: This subsection needs to be re-written. It did not contain specific information regarding participants.
Thank you for your detailed comments concerning the subsection 'Participants'. We have rewritten this subsection and added information regarding the participants, the classic and protracted variants of CLN3 disease and the (validation of the) UBDRS questionnaire. The patient files contain information that was collected by care professionals as common procedure (i.e. registration of diagnosis and progress/deterioration) and was kept safe in the registration system of Bartiméus (open to the concerning care professionals only). We have asked permission to the parents for using these files for our research and ethical approval for the study was granted by The Scientific and Ethical Review Board (VCWE), which has been described in the subsection 'Procedure'. A research proposal has been written before the study started, which was granted to us by The Netherlands Organization for Health Research and Development (ZonMw).
Line 122-129: Relocate this information in the corresponding subsections based on STROBE guidelines (STREGA if necessary).
We have moved this information to the subsection Participants, following the JARS-Qual reporting standards.
Line 130-152: "Consensus" of co-authors only? How is this valid in the light of the scientific community? Could the authors please explain why did not perform a Delphi exercise or at least a content validity analysis? How did you measure "reliability"?
Reliability and validity cannot be used in the same way in qualitative research as they can be used in quantitative research (Golafshani, 2003). In qualitative research, the consistency of the data will be achieved by examination of the steps of the research (such as raw data, data reduction process, etc.), for example with an audit procedure. The interrater reliability, a measure that indicates how much two researchers agree when they code the same data, is recommended as good practise in qualitative research and can be used to assess the robustness of the coding frame (O'Connor & Joffe, 2020). In our study, the interrater reliability was estimated by calculating the percentage of agreement between two coders. This is the percentage of data units on which coders agree.
Golafshani, N. (2003). Understanding reliability and validity in qualitative research. The qualitative report, 8(4), 597-607.
O’Connor, C., & Joffe, H. (2020). Intercoder reliability in qualitative research: debates and practical guidelines. International journal of qualitative methods, 19, 1609406919899220.
Line 146-165: These are the second and third steps of the "Framework analytic procedure"? English editing is needed to improve readability. Too long paragraphs make it difficult to understand clearly the performed procedures.
Line 166-174: This should be relocated as part of the "Data analysis".
Line 206-208: Not clear. Please elaborate on the analytical workflow.
Line 208-209: Delete.
We have rewritten the subsections Procedure and Data Analysis to make it better understandable, thereby taking into account your points of feedback.
RESULTS
This section needs to be improved. Too much text makes it difficult to interpret findings. Please follow STROBE guidelines and support your findings with data from the current study.
We improved the Results section, based on your specific comments below (see also our reactions below). The results of the qualitative analysis follow the structure of the overarching questions listed in the section Data analysis. By improving the section of Data analysis, we hope that the Results section is now also easier to read.
Line 211: all clients? They were not explained in the METHODS section. Only 7 clients? Where do they come from?
In total, we had 11 participants. However, from only 7 clients, UBDRS-data were available (see Table 1). We have tried to make this more clear in the text.
Line 217-218: Difficult to follow. What do the authors mean by "overview"? This is part of the Framework analysis? If so, how can we as researchers validate the objectivity of the results?
The overviews indeed follow from the framework analysis. We have rewritten the section Procedure and Data analysis to elaborate on the Framework analysis and to explain better which steps we followed in the analysis. For the validity and objectivity of the results, please see my answer to one of your previous questions above.
Line 227-228: "However, no typical 227
combinations of problems and no typical development of problems could be identified": How do you support this statement based on your findings? Lack of reproducibility.
In qualitative research, results such as overviews or relations are usually supported with examples from the data. In the case that no relation is found, no examples can be given.
Line 230-272: Causation is beyond the scope of this study design. Authors should avoid any attempt to explain causation if any molecular, mechanistic, or robust statistical modeling (even with AI or ML procedures) were performed.
We agree that we cannot be entirely sure of the possible causes for the emotions and behaviour that we listed. We slightly changed the wording in order to be more precise. We would like to add, however, that why- and how-questions are common in qualitative research and that it has even been argued that causality has a place in qualitative research (Donmoyer, 2012)
Donmoyer, R. (2012). Attributing causality in qualitative research: viable option or inappropriate aspiration? An introduction to a collection of papers. Qualitative Inquiry, 18(8), 651-654.
Line 276-279: Please report those correlation outcomes! Please elaborate differences between brothers in terms of the genetics analyses.
The hypothesized relation between the severity of behavioural problems and environmental or situational factors has only be assessed qualitatively. We understand that the term 'correlation' is misleading here and we have therefore changed it to 'coincide'. We apologize for the confusion. Concerning the genetics of the two brothers: the only genetic information we have about them is that they both were identified with the classic form of CLN3 disease (see Table 1).
DISCUSSION
This section is far from objectivity and is non-supported by the findings of this study based on the lack of reproducibility and validity of the data.
Perhaps a scoping review would be a better form to use the information collected by the authors given the lack of standardization of the current study - which is highly expected and required in a retrospective cross-sectional study.
The main part of our study is qualitative, and for qualitative research validity and reproducibility cannot be easily measured, as we explained before. We did our best to follow the guidelines for qualitative research (such as calculating interrater reliability) and discuss our qualitative findings in the Discussion. We have added some text to the discussion to elaborate on and compare our findings. Doing a scoping review on the literature of our research topic, might be a good idea as well, however, it cannot be performed on the data (patient files and interviews) that we used.
CONCLUSIONS
These statements should be based on your findings according to the limits of the study aim. Generalities are not accepted as conclusions.
We agree that the conclusions were not connected well enough to our study. We have rewritten this part.
We thank you again for your careful review and suggestions. We believe that by revising our manuscript along the lines of your feedback, the method of analysis (with its implications about validity and reproducibility) is now more clearly reported.
Reviewer 3 Report
REVIEW of the manuscript titled:
ijerph-1655194
Towards Understanding Behaviour and Emotions of Children with CLN3 Disease (Batten Disease): A Retrospective File Research
The approach to the Baten disease from the perspective of recurrent losses and disruption of the family system, although not a novelty, is far from finding the complete and most appropriate solutions. In this sense, the authors use updated references, including from 2022.
Please find some of my observations.
The title is specific, and descriptive, including the keywords of the research. The abstract is a concise summary of the research conducted – the topic, the purpose of the methodology and the conclusions of the research.
The Introduction presents general information on the subject, conclusions of the previous studies in correlation with the situation in the investigation area. The authors exploit a gap in previous research – the research is oriented towards practice and aims at a better understanding of the behaviour of children with CLN3 disease, emotional problems and the necessary support to be given to children but also to parents.
The applied methodology and also the data analysis method to solve the problem was presented. The results and discussions of the study show that significant advances have been made. The authors also present some limitations of the study related to the behaviour and emotions related to the stage of the disease.
A topic for future research is also proposed – in the support programs for children affected by this disease, the whole family, including the siblings of these children deserves special attention.
I agree with the form presented for publication.
- The Unified Batten Disease Rating Scale (UBDRS) is a reliable instrument that is useful for monitoring the diverse clinical findings seen in Batten disease.
- For this study, the rules of scientific and ethical evaluation were observed.
- The article writing was clear, respects the classic structure and is presented correctly.
- Batten’s disease has several forms, which have similar manifestations but differ in the patient’s age and the severity of the disease at onset. A weak point may be the non-uniformity of the analyzed group, at least from the perspective of age.
- From the point of view of temporality, this study is retrospective. From the point of view of validity (correctness), the author should have relied on the results of a meta-analysis of clinical trials to demonstrate the hypotheses.
- The chosen subject addresses a major problem that it does not solve. However, it makes observations that may be important for future research.
Overall, I believe that although it is not a state-of-the-art study, it may be considered for further research.
Author Response
Dear reviewer,
Thank you very much for your evaluation of our study and valuable points of feedback. We agree that taking the behavioural perspective does not solve the disease. However, we find it important, until CLN3 disease can be cured, to support the life of these children and their parents as well as possible.
We agree with you on the weak point of the non-uniformity of the analysed group, and added this to the discussion.
With respect to your comment on the validity of the study, we would like to point out that the main part of our study is a qualitative analysis in which explorative questions without clear hypotheses are common. To be clearer and explain this aspect we revised the Procedure. In the manuscript, track-changes have been used to indicate changes.
Thank you again for your review.
Round 2
Reviewer 1 Report
The manuscript has undergone a serious revision and generally meets the high requirements of the journal. In general, it can be published. There are only two wishes for authors and editors at their discretion.
Throughout the text there are formulations such as "illness causes emotional and behavioral changes" (p2 line68), which to a certain extent creates an incorrect idea of a direct direct relationship, In fact, such changes are often mediated and are the result of subjective experiences due to a severe physical condition, which, in turn, causes changes in the behavioral sphere. It would be better to replace "causes" with "leads to" or something like that.
The conclusion states "problematic behavior and emotions" (p12 line523) - despite the fact that it is obvious what the authors mean, it is probably worth refraining from such evaluative epithets - emotions are not problematic at all, and behavior in the context of this study is considered as a target of the negative influence of the disease, and not an antisocial action.
Author Response
Dear reviewer,
Thank you for these valuable remarks. We fully agree with both of your comments and thank you for additional explanation. Indeed, it is important to rephrase several sentences so as to be more careful and specific in what we mean. We have rephrased sentences containing the word 'causes' and used 'leads to' as a replacement most of the times. We chose to change the word combination 'problematic behaviour and emotions' into 'behavioural and emotional symptoms' most of the times. A few times we used the words 'typical behaviour and emotions' and 'disturbed behaviour and emotions' at different places in the text where this was most applicable. In some cases where we cited articles that did use the word 'problems', we kept this word in our manuscript as well. In the document, we have used Track Changes such that the changes that we made can be easily found.
Thank you again for your dedicated review.
Reviewer 2 Report
Dear authors "Towards Understanding Behaviour and Emotions of Children with CLN3 Disease (Batten Disease): patterns, problems and support for child and family",
Thanks for re-submitting after having partially addressed my previous comments/suggestions. The scientific soundness of the manuscript has increased a little bit; notwithstanding, please respond to each of my comments/questions to revise this manuscript in further detail.
- Some of your responses are repeated but the questions are not the same.
- The manuscript's extension is still excessive.
- Not sure about the suitability of attributing the features of qualitative research and differences from quantitative methods (https://doi.org/10.1007/s11133-019-9413-7) to several of your responses.
- Readers would benefit from a clear explanation of "overviews" meaning and interpretation. I understand that coding is part of this but this needs to be improved for readability.
Author Response
Dear reviewer,
We are pleased that we have the possibility to further address your comments/questions. We respond to each of your comments below.
- Some of your responses are repeated but the questions are not the same.
We understand this to be related to your previous questions about the research methods, addressing, among others validity and reliability. In our previous reply we explained that validity and reliability are different concepts in qualitative research and described how this is handled. We looked back into your list of previous questions and saw that we may have not fully answered your question on objectivity of the results. We apologize for this. Objectivity in the purest sense, the idea that a truth exists outside any investigation or observation, does not exist in qualitative research. However, by having multiple coders, we try to avoid bias and derive at results that are as objective as possible with this method. In our study, four coders were present.
- The manuscript's extension is still excessive.
This indeed is a difficult point to address. In the previous revision another reviewer suggested to include more references to embed our research more clearly in the already existing literature. We tried to be shorter in the text but we needed to provide more information as well. At this point it is difficult to remove information. We hope for your understanding.
- Not sure about the suitability of attributing the features of qualitative research and differences from quantitative methods (https://doi.org/10.1007/s11133-019-9413-7) to several of your responses.
Thank you for pointing out this interesting study focusing on the similarities between qualitative and quantitative research. The article gives a definition of what qualitative work is, according to them, and even proposes four characteristics. However, the question of whether a qualitative study contains all four characteristics can only be answered by discussing the method and results in terms of these characteristics. It will be interesting to follow how this proposed evaluation continues, so thank you again for highlighting this article.
- Readers would benefit from a clear explanation of "overviews" meaning and interpretation. I understand that coding is part of this but this needs to be improved for readability.
We agree that the term 'overview' could be explained better for readability. To be clearer we have added information to the sections Procedure and Data analysis. This explains more clearly how the overviews or summaries resulted from the analysis. See pages 4-5.
Thank you again for your review. Your feedback has contributed much to the improvement of the manuscript, for which we are very grateful.